# Competing protein-protein interactions regulate binding of Hsp27 to its client protein tau

Rebecca Freilich[1], Miguel Betegon[2], Eric Tse[2], Sue-Ann Mok [1], Olivier Julien [1], David A. Agard[2], Daniel R. Southworth[2], Koh Takeuchi [3] & Jason E. Gestwicki [1]

Small heat shock proteins (sHSPs) are a class of oligomeric molecular chaperones that limit protein aggregation. However, it is often not clear where sHSPs bind on their client proteins or how these protein-protein interactions (PPIs) are regulated. Here, we map the PPIs between human Hsp27 and the microtubule-associated protein tau (MAPT/tau). We find that Hsp27 selectively recognizes two aggregation-prone regions of tau, using the conserved β4-β8 cleft of its alpha-crystallin domain. The β4-β8 region is also the site of Hsp27–Hsp27 interactions, suggesting that competitive PPIs may be an important regulatory paradigm. Indeed, we find that each of the individual PPIs are relatively weak and that competition for shared sites seems to control both client binding and Hsp27 oligomerization. These findings highlight the importance of multiple, competitive PPIs in the function of Hsp27 and suggest that the β4-β8 groove acts as a tunable sensor for clients.

[1] Department of Pharmaceutical Chemistry, University of California San Francisco, San Francisco, CA 94158, USA. [2] Department of Biochemistry and Biophysics, University of California San Francisco, San Francisco, CA 94158, USA. [3] Molecular Profiling Research Center for Drug Discovery, National Institute of Advanced Industrial Science and Technology (AIST), Tokyo 135-0064, Japan. Correspondence and requests for materials should be addressed to J.E.G. (email: jason.gestwicki@ucsf.edu)

Molecular chaperones maintain cellular protein home-ostasis (proteostasis)[1]. Among these chaperones, the small heat shock proteins (sHSPs) play a key role by preventing the aggregation of partially unfolded proteins[2–4]. Specifically, sHSPs are thought to maintain their client proteins in a soluble, folding-competent state for subsequent processing by ATP-dependent chaperones[5,6], such as Hsp70. In this way, the sHSPs act as sentinels of protein unfolding, especially in response to stress or conditions that would promote protein aggregation.

Hsp27 is a broadly expressed member of the sHSP family, which prevents aggregation of a large number of putative clients[7]. As a consequence of these interactions, Hsp27 has been impli-cated in many diseases, including neurodegeneration[8–10]. Despite its important roles, the molecular mechanisms of Hsp27 function remain mysterious. Like all sHSPs, Hsp27 contains a highly conserved α-crystallin domain (ACD) flanked by disordered N- and C-terminal domains (NTD and CTD) (Fig. 1a). The ACD has an anti-parallel β-sandwich fold and cross β-sheet interactions between two of these domains mediate the dimerization of sHSPs[11,12]. These dimers are then assembled into larger species (up to ~30 mers) through a series of distinct PPIs that involve different regions of the ACD, as well as the NTD and CTD. The best characterized of these oligomer-stabilizing PPIs is the one between IXI motifs in the CTD and the β4–β8 groove of the ACD[12–15]. This interaction involves binding of the linear, dis-ordered IXI motif into a shallow groove between β-sheets 4 and 8. The IXI interaction with β4–β8 is important in homo-oligomer formation but it can also facilitate heterodimer formation between different members of the sHSP family[16]. Separate PPIs involving the NTD are also thought to contribute to oligomer formation[16–18], but the specific interaction sites are not known. However, Hsp27's NTD is clearly important because three phosphorylation sites in that region regulate oligomer assembly[19].

One of the major roles of Hsp27 is to prevent aggregation of its client proteins. However, it is not yet clear where Hsp27 binds to clients or how it stabilizes them. Work in other sHSPs, such as α-crystallin[20], yeast Hsp42[5], and plant Hsp18[21–23] has suggested that different surfaces can be used to engage clients. It has been shown that the highly conserved ACD is sufficient to prevent aggregation of certain clients[12,20], while the NTD is necessary for others[20,24–27]. However, the factors that determine what types of clients are bound and what parts of Hsp27 are involved (i.e., ACD, NTD, and CTD) remain ambiguous. One of the key issues is that the natural clients of the sHSPs are often not known. Model proteins, such as firefly luciferase, citrate synthase and malate dehydrogenase, have been used to study mechanisms of sHSP function[5,22,28], but biophysical information on native interactions is sparse[7,29,30].

In this study, we characterized the interactions between Hsp27 and microtubule-associated protein tau (MAPT/tau). We selected tau for these studies because Hsp27 has been shown to regulate tau's aggregation in vitro and in vivo[31,32], making it one of the few well-validated, natural clients. Moreover, tau is an intrinsi-cally disordered protein, a feature that removes the complication of considering both folded and unfolded states. Finally, tau is known to aggregate in a number of progressive, untreatable neurodegenerative disorders[33], including some forms of Alzhei-mer's disease (AD), so mechanistic information about how Hsp27 acts on it would be of biomedical interest. Indeed, overexpression of Hsp27 has been found to be partially protective in animal models of tauopathy[31].

Using a combination of biophysical techniques, we characterize Hsp27's interactions with tau. We find that Hsp27 binds two sites on tau (PHF6 and PHF6*) that are well known to be required for aggregation. The location of these interactions helps explain the strong anti-aggregation activity of Hsp27 and suggests that the chaperone may be tuned to prefer the most problematic regions in tau. On the other side, we find that Hsp27 uses both its NTD and β4–β8 groove to bind tau. The involvement of the β4–β8 groove was unexpected and interesting because, as mentioned above, this region is also known to bind IXI motifs within the sHSP oligomer. Thus, this result suggests that competition between client and IXI might limit chaperone function. Indeed, we find that Hsp27–Hsp27 interactions, mediated by the IXI motifs, seem to partially counteract tau binding. We speculate that this set of competing PPIs may normally hold Hsp27 in a latent, inactive oligomer prior to accumulation of sufficient levels of misfolded client to overcome the self-limiting interactions. Finally, we find that ACD binding alone is not sufficient to prevent tau aggregation and that additional contacts in the NTD are required. Together, these results suggest that, at least for tau, Hsp27's NTD plays a major role in chaperone function, while the β4–β8 groove serves as an important client sensor. This model advances our understanding of sHSP structure-function and suggests potential ways of targeting Hsp27 for drug discovery.

## Results

**Hsp27's β4/β8 groove is a PPI hot spot**. Because the ACD has been shown to have chaperone activity for some clients[20], we first measured the interaction of [15]N-labeled Hsp27 ACD with tau by HSQC NMR[11]. In this experiment, we utilized two isoforms of tau (Fig. 1a and Supplementary Figure 1): the full-length splice variant 0N4R tau, which is commonly expressed in adult brain and K18, a construct containing just the four microtubule-binding repeats (MTBRs) that is known to contain the binding sites of other chaperones[34,35]. We found that both tau isoforms caused chemical shift perturbations (CSPs; >0.4 ppm) within the hydrophobic β4–β8 groove of the ACD, in addition to sporadic shifts along the dimer interface (Fig. 1b). The β4–β8 groove is a conserved binding site for IXI motifs[14], as well as the Hsp70 co-chaperone BAG3[36] and some other clients[20]. Moreover, a recent study also suggested that tau binds to this region of Hsp27's ACD[32]. To better understand how tau and the IXI motif (which is isoleucine, proline, valine (IPV) in Hsp27) might compete for the same region, we first synthesized the 9-mer peptide of Hsp27 ([176]EITIPVTFE[185]) and confirmed its binding site in the HSQC assay (Fig. 1c). We found that the peptide caused peak broad-ening, indicating that it is in an intermediate exchange. As expected, the affected peaks were localized to the β4–β8 groove, confirming that both the IPV motif and tau bind the same region.

Next, we wanted to understand the relative affinity of the IPV motif and tau protein for the β4–β8 groove. First, we used isothermal calorimetry (ITC) to measure the affinity of Hsp27's IPV peptide for the β4–β8 groove in the purified Hsp27 ACD (Fig. 1d and S2A). We found that the native Hsp27 IPV sequence had a weak affinity for ACD (>25 μM). This weak affinity for the isolated peptide made it difficult to identify the residues that might contribute to affinity. Therefore, we swapped the penultimate phenylalanine residue for a histidine (EITIPVT**H**E), which is the identity of this position in the tighter binding motif of the co-chaperone BAG3. As expected, this change improved the affinity of the chimeric peptide to 11.5 ± 0.43 μM (see Fig. 1d). Then, in the context of that His-containing sequence, we examined the impact of mutating other residues that might contribute to affinity using alanine scanning. In these studies, we were particularly intrigued by the fact that Hsp27's motif contains an unusual, extended IXIXV sequence (ITIPV; see Fig. 1d). However, alanine scanning revealed that the isoleucine (Ile179) and valine (Val181) of the canonical IPV are the major contributors to affinity; mutating either one of these residues completely ablated the interaction ($K_d$ > 25 μM). On the contrary,

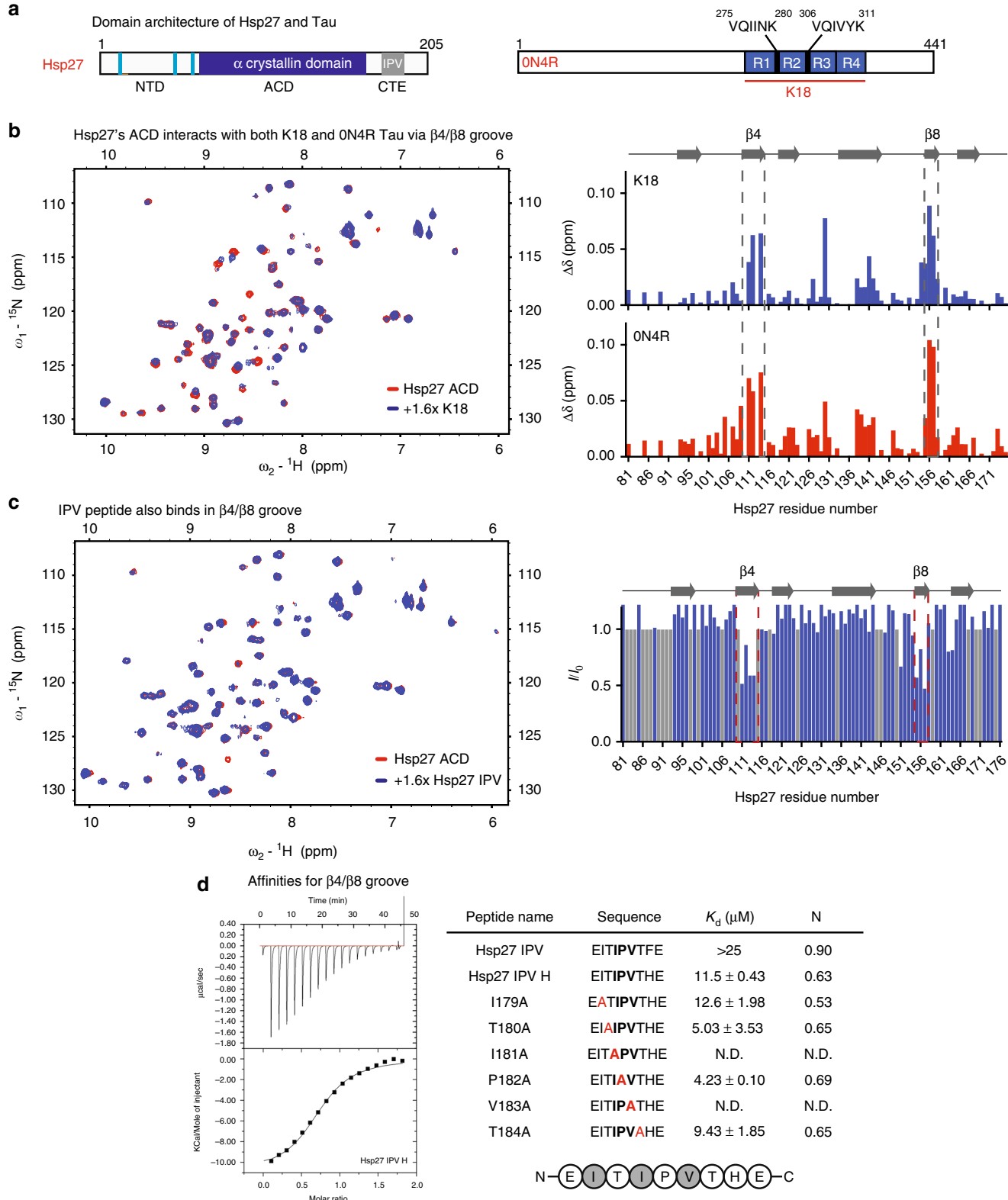

**Fig. 1** Hsp27's β4/β8 groove is a PPI hot spot for both client- and self-interactions. **a** Domain architecture of Hsp27 and Tau isoforms. **b** Left, HSQC spectra of $^{15}$N Hsp27 ACD alone (150 μM, red) or in the presence of 250 μM K18 (blue). Right, chemical shift perturbations in ACD upon binding of 250 μM K18 (top) or 0N4R (bottom). **c** Left, intensity ratios upon binding of Hsp27 IPV peptide, with unassigned residues shown in gray. Right, HSQC spectra of $^{15}$N Hsp27 ACD alone (150 μM, red) or in the presence of 250 μM Hsp27 IPV peptide (blue). **d** Isothermal calorimetry of Hsp27 ACD with IPV-derived peptides. Left, representative ITC curve for Hsp27 IPV H peptide. Right, table of affinity values. ND, no detectable binding. Values are represented as mean ± standard error of the mean (SEM) determined from a minimum of three independent experiments. Bold letters highlight the mutated residue

mutating either X residue in the IXIXV motif (EIA**TIPVTHE** or EITI**A**VTHE) modestly improved, rather than weakened, affinity ($K_d \sim 5\ \mu M$). The ITC titrations indicated that all of the peptides have a binding stoichiometry of between 0.5 and 1, suggesting that the peptides do not always occupy all available β4–β8 binding sites on the ACD dimer. Thus, from these experiments, we concluded that the native IPV peptide of Hsp27 (EITIPVTFE) has a weak affinity for the β4–β8 groove (>25 μM) and that Ile179 and Val181 are important for the interaction.

Given that the IPV peptide binds weakly, we wondered whether tau might have a stronger affinity for the Hsp27 ACD. However, we expected this interaction to be quite weak, because of previous NMR titrations[32] and, consistent with this idea, we were unable to achieve saturable binding of K18 tau to Hsp27's ACD by a competitive ELISA (estimated $IC_{50} > 100\ \mu M$; Supplementary Figure 3A), Likewise, solubility prevented accurate measurement of this affinity by ITC ($K_d > 100\ \mu M$; Supplementary Figure 3B). Thus, we concluded that both IPV peptides and tau bind at the β4–β8 groove with weak affinity.

**Hsp27 binding to tau can be obscured by self-interactions**. Next, we wanted to study the other side of the interaction and learn where Hsp27 binds on tau. Because our earlier study suggested that K18 and 0N4R tau bind similarly to Hsp27's ACD (see Fig. 1), we focused on the shorter K18 region as likely to contain the major site(s). Accordingly, we used $^{15}$N-labeled K18 in HSQC NMR experiments to map putative interactions with unlabeled Hsp27 ACD. We found that addition of Hsp27's ACD caused peak broadening in two remarkably discrete locations (Fig. 2). These binding sites include the well documented PHF6 ($^{275}$-VQIINK-$^{280}$) and PHF6* ($^{306}$-VQIVYK-$^{311}$) regions, which are known to be required for tau aggregation[37]. To confirm whether the β4–β8 groove was involved in this binding, we purified an Hsp27 ACD mutant, L157A. This residue was previously identified as being important in IXI interactions[14] and we confirmed that mutation blocks binding to BAG3 (Supplementary Figure 4). Surprisingly, we found that the L157A mutation ablated the interaction with the first motif, but not the second (Fig. 2), suggesting that the binding modes of the two peptide regions may not be equivalent.

In order to study the two binding sites in more detail, we carried out a titration analysis using a TROSY scheme with better chemical shift resolution[38]. Titration of Hsp27 ACD into $^{15}$N-labeled K18 tau confirmed that the main binding interaction occurs at the two PHF6 and PHF6* motifs, with minor interactions at residues 292–297 and 349–353 (Supplementary Figure 5A). Consistent with the importance of these motifs, deletion of residues 277 and 278 (K18 Δ277–278) in the middle of VQIINK limited binding to PHF6, while deleting residues 308 and 309 (K18 Δ308–309) had the same effect as PHF6* (Supplementary Figure 5B). Likewise, deleting both sets of residues (K18 double mutant) entirely abolished the interaction with Hsp27's ACD. Consistent with the studies above, we found that the affinity of Hsp27 ACD for K18 tau was weak and unsaturable by NMR titrations (Supplementary Figure 3C). Likewise, titration of the 6-mer PHF6* peptide (VQIVYK) was unable to produce saturable chemical shift changes in the $^{1}$H–$^{15}$N ACD spectra (Supplementary Figure 3D), supporting the idea that the isolated interactions are weak. Together, these studies showed that Hsp27 ACD and tau interact using at least two regions: the β4–β8 groove of the ACD and the PHF6 and PHF6* motifs of tau.

The NTDs of sHSPs have also been implicated in recognition of various model clients. To determine whether it is involved in tau binding, we purified Hsp27's NTD and tested its binding to $^{15}$N K18 by HSQC. This experiment revealed broadening in a large number of residues across the K18 sequence (Fig. 2). Compared to the discrete ACD interaction, the NTD contacts seemed to involve multiple parts of K18 tau, including the PHF6 and PHF6* aggregation motifs, as well as broad regions corresponding to residues 246–268 and 342–361.

The evidence so far suggested that Hsp27 interacts with tau using both its ACD and NTD. We hypothesized, therefore, that the full-length Hsp27 protein, containing both the NTD and the ACD, might have tight affinity due to multivalent interactions. However, when we tested binding of full-length Hsp27 to K18 tau (Fig. 2 and Supplementary Figure 6), we observed no broadening or CSPs. This result is consistent with an alternative model in which the Hsp27–Hsp27 interactions, mediated by the IPV motifs locking together with β4–β8 grooves in adjacent ACDs, largely hides this PPI surface from interacting with tau. Indeed, previous structural studies[15] have shown that Hsp27's IPV motifs engage in extensive PPIs with itself in the context of large oligomers. Based on these results, we speculate that the weaker interactions of tau cannot, under these conditions at least, compete with Hsp27's IPV motifs in the context of larger oligomers. In other words, although both tau and IPV peptide bind Hsp27's ACD weakly as isolated peptides (see Fig. 1), the high local concentration of IPVs in an intact Hsp27 oligomer seems to create an avidity effect that outcompetes binding to monomeric tau. Moreover, our observations also suggest that even the NTDs must be restricted from interacting with monomeric tau in the context of full-length Hsp27. We conclude that the NTDs, which bind to tau in isolation, are somehow masked in the context of Hsp27 oligomers (see below).

**Hsp27's NTD plays a functional role in chaperoning tau**. So then, how does Hsp27 become activated as a chaperone? To explore this question, we first focused on the phosphorylations of the NTD because these have been shown to reduce oligomer size and increase chaperone activity in other systems[19,24]. Thus, we reasoned that phosphorylation of the NTD might partially release its availability for binding to tau. This activation might occur through multiple mechanisms: (a) increasing the availability of unoccupied β4–β8 grooves by reducing oligomer size and freeing the self-IPV motifs and/or (b) increasing the display of available NTDs.

To explore these ideas, we first measured the ability of Hsp27 to suppress aggregation of tau in vitro, as measured by thioflavin T (ThT) fluorescence. Here, we used the suppression of tau aggregation as a model chaperone function. Before starting these studies, we first confirmed previous observations[31] that full-length Hsp27 prevents aggregation of both 0N4R and K18 in the ThT assay (Fig. 3a), significantly increasing the lag time and slowing the elongation rate. The chaperone also sometimes produced an increase in the apparent maximal level of fluorescence signal, which may occur because it is partially incorporated into the insoluble fraction (Supplementary Figure 7). Why can WT Hsp27 suppress tau aggregation if its IPV motifs are already engaged in self-interactions? Presumably, Hsp27's ACDs and NTDs bind to tau under these conditions because the high local concentration of the client, in the context of a growing fibril or earlier intermediate[32], outcompetes the self-limiting contacts. Although this possibility is difficult to directly measure, this idea could provide a way for the chaperone to remain off until clients become damaged.

Using the ThT assay, we next explored the roles of NTD and its phosphorylation. We found that deletion of the NTD (ΔNTD) ablated Hsp27's ability to delay tau aggregation, showing that this region is essential to chaperone function. Consistent with this

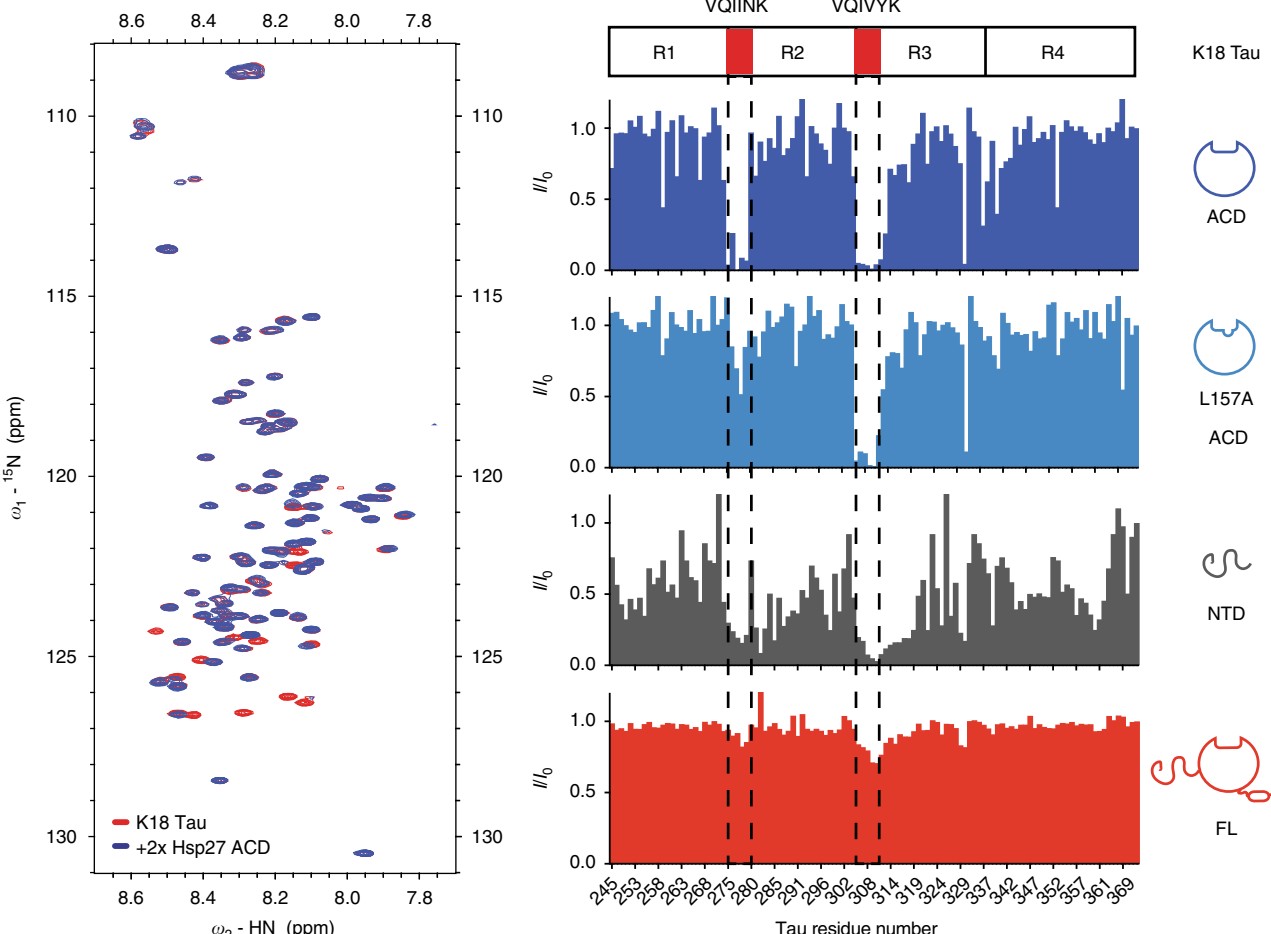

**Fig. 2** Individual domains of Hsp27 bind to K18 tau, but full-length Hsp27 is masked. Left, HSQC spectra of $^{15}$N K18 alone (50 μM, red) or in the presence of 100 μM ACD (blue). Results are representative of experiments performed in triplicate. Right, intensity ratios upon binding of listed Hsp27 constructs at a 2:1 molar ratio

result, the ACD alone had no effect on tau aggregation, despite the fact that it would be predicted to (weakly) bind to the PHF6 and PHF6* motifs. The ACD is known to have chaperone activity in other amyloid-prone client systems[20], but it does not seem to be sufficient for tau.

Then, we explored the effect of NTD phosphorylation on binding to tau. Specifically, we used a pseudo-phosphorylation mutant, termed 3D, in which three key serine residues (S15, S78, and S82) are replaced with aspartate as a phospho-isostere[31]. We first confirmed that oligomers of Hsp27 3D are more heterogeneous and generally smaller than oligomers of WT Hsp27, as measured by negative-stain EM and SEC-MALS (Fig. 3b), but with roughly equivalent melting temperature (Supplementary Figure 8A). In both the intensity-based HSQC (Fig. 3c) and the chemical shift-based, TROSY-HSQC (Supplementary Figure 6) platforms, we found that Hsp27 3D binds to K18's aggregation motifs (PHF6 and PHF6*), while WT does not. Also, we found that Hsp27 3D retains the ability to prevent tau aggregation by ThT assays (Fig. 3d), although we were not able to confidently determine whether WT or 3D Hsp27 were the superior chaperone (Supplementary Figure 7). Together, these results suggest that reducing oligomer size, through pseudo-phosphorylation of the NTD, partially releases the internal, self-limiting PPIs and activates tau binding and anti-aggregation activity.

**The β4/β8 groove regulates sHSP oligomerization and function.** These results suggest a model in which the NTD is a major domain for chaperone function, at least for tau, while the β4–β8 groove of the ACD is a hot spot for regulatory PPIs. To further explore the role of the β4–β8 groove as a sensor, we wondered what effect weakening the self IPV–ACD interactions might have. Based on the alanine scanning and recent work[32], we mutated the isoleucine and valine residues in the IPV to glycine (Hsp27 GPG). Hsp27 GPG has an altered oligomeric structure as compared to WT (Supplementary Figure 8B) and, as expected, it has a lower apparent melting temperature, as measured by differential scanning fluorimetry (DSF) (Fig. 4a).

To understand whether the Hsp27 GPG mutant created additional opportunities for tau binding in the β4/β8 grooves, we first measured its affinity for IPV peptides by ITC (Supplementary Figure 2B). In this experiment, we again used the His-containing IPV peptides as a surrogate for binding site availability because they have tighter affinity than tau or the native sequence, so are more accurate to measure. As expected, we found that WT Hsp27 does not bind IPV peptide (EITIPVTHE; Fig. 4b), likely due to the high occupancy of the β4–β8 grooves. However, Hsp27 GPG bound to IPV with an affinity of ~10 μM, similar to that of an isolated ACD (see Fig. 1d). This result suggests that easing the internal competition can, indeed, partially expose β4–β8 grooves and promote binding

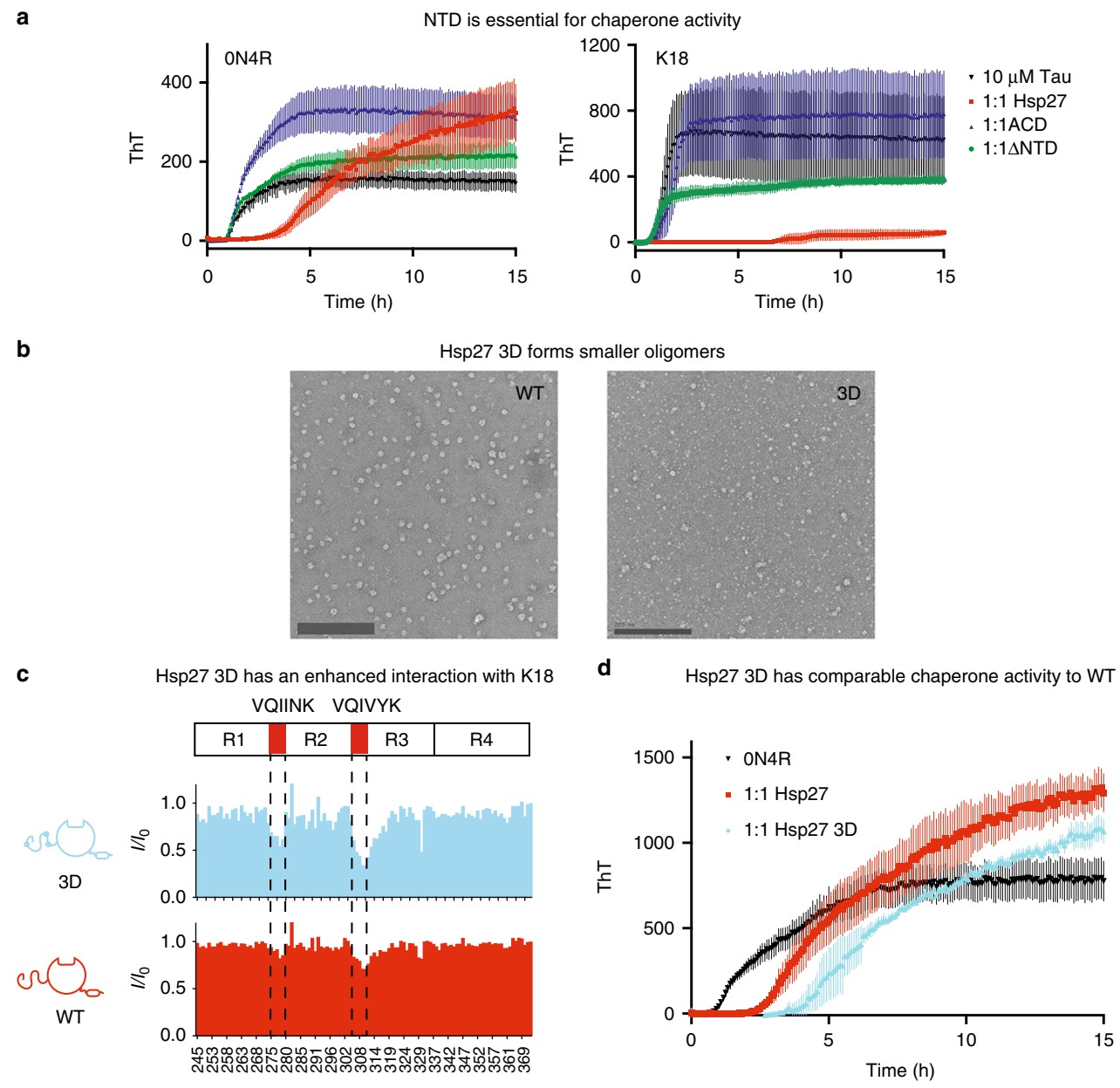

**Fig. 3** Hsp27's NTD plays an important role in chaperoning tau. **a** Representative tau aggregation curves in the presence of truncated Hsp27 constructs. Aggregation of 0N4R (left) or K18 (right) in the presence of listed chaperones at a 1:1 molar ratio (10 μM). Data points are the mean ± SEM of 3 technical replicates. RFU, relative fluorescence units. **b** Left, Negative-stain EM images of WT (left) and 3D (right) Hsp27. Images are representative of a minimum of 12 random fields. Scale bar is 200 nm. Right, SEC-MALS trace of Hsp27 WT and 3D (40 μM). **c** Intensity ratios of 50 μM $^{15}$N K18 in the presence of of 100 μM Hsp27 WT or 3D (2:1 molar ratio). **d** Representative 0N4R aggregation curves in the presence of Hsp27 WT or 3D at a 1:1 molar ratio (10 μM). For information on why the amplitude of the signal may be increased in the presence of chaperones, see Supplementary Figure 7C. Results are representative of independent experiments performed in quadruplicate (see Supplementary Figure 7A). Each individual experiment was performed in triplicate and error bars represent SEM

to client. However, Hsp27 GPG had a reduced binding stoichiometry ($n = 0.2$), indicating that not all β4–β8 grooves are available. To further probe the role of oligomeric interactions in the availability of β4/β8 grooves, we deleted the NTD from Hsp27 WT (ΔNTD) and Hsp27 GPG (ΔNTD GPG). These proteins were intended to release the NTD contacts entirely and create smaller oligomers. Indeed, the ΔNTD GPG construct is a dimer (Supplementary Figure 8C), consistent with the partial loss of both the NTD and IPV contacts. However, ΔNTD results in a

heterogeneous mixture of 3–5 mers (Supplementary Figure 8C), likely because of the residual IPV–ACD contacts. Despite these slight differences, both ΔNTD and ΔNTD GPG had similar affinities for the IPV peptide (EITIPVTHE; 11.4 and 14.3 μM, respectively). Thus, one model to explain this result is that more β4–β8 grooves may be available for binding in the absence of NTD. Finally, we wondered whether Hsp27 3D would contain a greater number of free β4/β8 grooves because of its smaller oligomeric size. However, Hsp27 3D bound poorly to the

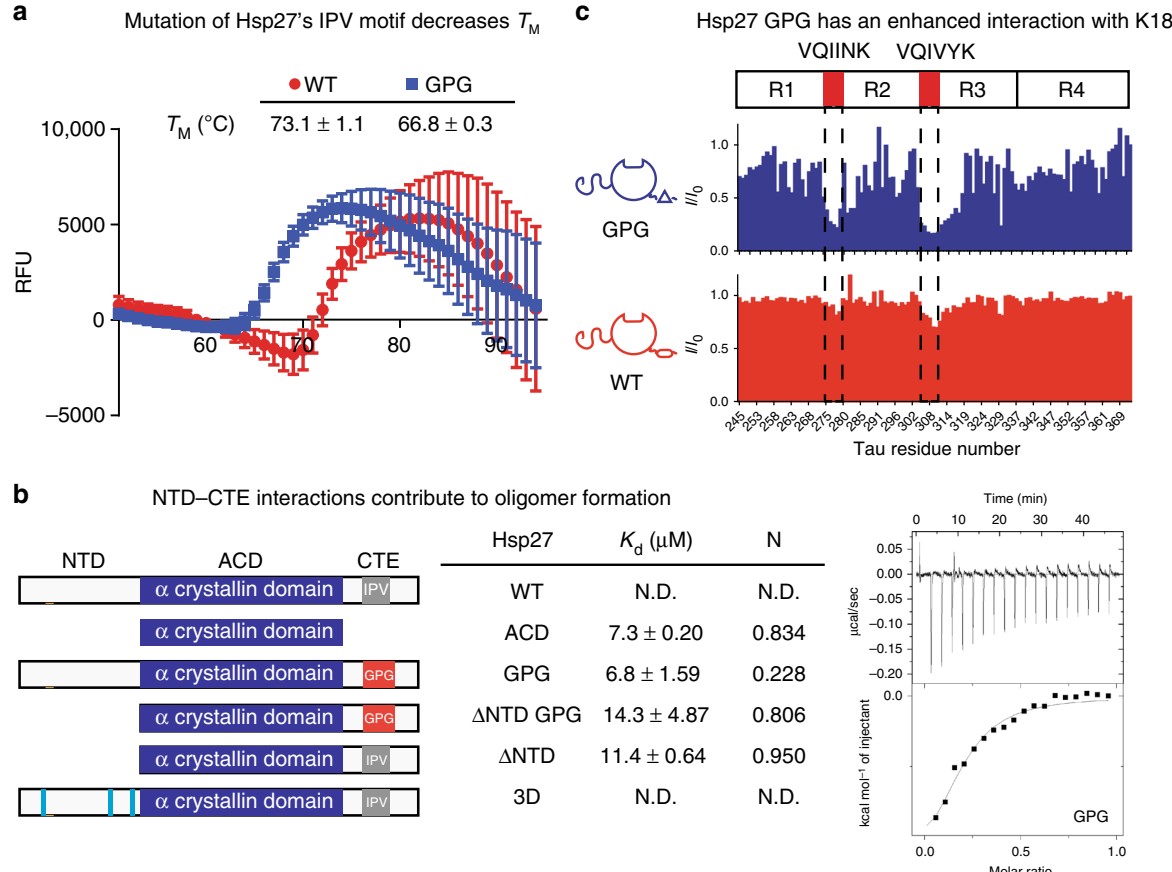

**Fig. 4** The β4/β8 groove is a regulatory site of sHSP oligomerization and function. **a** Differential scanning fluorimetry melting curves of Hsp27 WT and GPG. Data points are the mean ± SEM of three technical replicates. Listed melting temperatures are the mean ± SD of three independent experiments. **b** ITC binding experiments of optimized IPV peptide with listed constructs of Hsp27. Left, cartoon representation of Hsp27 constructs used in this experiment. Middle, table of affinity and stoichiometry values obtained from ITC experiments. Values in table are mean ± SEM determined from a minimum of three independent experiments. Right, representative ITC curve of peptide binding to Hsp27 GPG. **c** Intensity ratios of $^{15}$N K18 in the presence of 100 μM Hsp27 WT or GPG (2:1 molar ratio)

His-containing IPV peptide ($K_d > 50$ μM), suggesting that there is still sufficient β4/β8 groove occupancy in this oligomer to exclude peptide. Thus, the number of available β4–β8 grooves seems to be a product of both internal competition for IPV motifs, as well as less well understood contributions from relative oligomer size and avidity effects.

After using the IPV peptides as a surrogate for available tau binding sites, we confirmed that Hsp27 GPG could interact with K18 tau by NMR. We found that Hsp27 GPG bound to K18 by both the intensity- and chemical shift-based NMR (Fig. 4c and Supplementary Figure 6), further highlighting the importance of competition between client and self in Hsp27.

**Dynamic PPIs are important for anti-aggregation activity.** These studies suggested that a series of competitive PPIs, driven by self- and client-interactions, as well as changes in phosphorylation and oligomerization, might coordinate Hsp27 function. This concept is in agreement with results from cryo-EM, mass spectrometry and SAXS studies of related systems, suggesting the importance of dynamics in sHSP oligomers[39–42] and sHSP-client interactions[23,28]. While it is difficult to accurately measure changes in Hsp27-tau dynamics during a chaperone cycle, we decided to cross-link Hsp27 and ask what effect this chemical modification would have on its ability to suppress tau aggregation. Specifically, we treated WT Hsp27 with gluteraldehyde and

selected a concentration of crosslinker that would produce samples with a similar oligomeric size to the untreated protein by SEC-MALS (Fig. 5a, Supplementary Figure 8D). Consistent with a role for dynamics, we found that the cross-linked sample (Hsp27 X) was severely compromised in suppressing aggregation of 0N4R or K18 tau (Fig. 5b). Although further work is needed, this finding is supportive of the idea that shifting PPIs regulate sHSP function.

**Discussion**
Previous work had shown that the ACD alone is sufficient to bind some clients but not others[12,20,43], suggesting that sHSPs may use different binding sites depending on the type of client. Here, we focused on mapping the interactions of Hsp27 with its native client, tau, which is important in AD and other neurodegenerative tauopathies[33]. We found that both the NTD and ACD of Hsp27 bind to the PHF6 and PHF6* aggregation motifs of tau, but that the accessibility of Hsp27's binding surfaces are limited in the full-length protein by self-interactions within the oligomer. Importantly, each of these individual PPIs was determined to be relatively weak in isolation (>25 to >100 μM). However, avidity effects at high local concentration, such as when tau begins to aggregate (see Fig. 3), seems to tune these relative binding affinities.

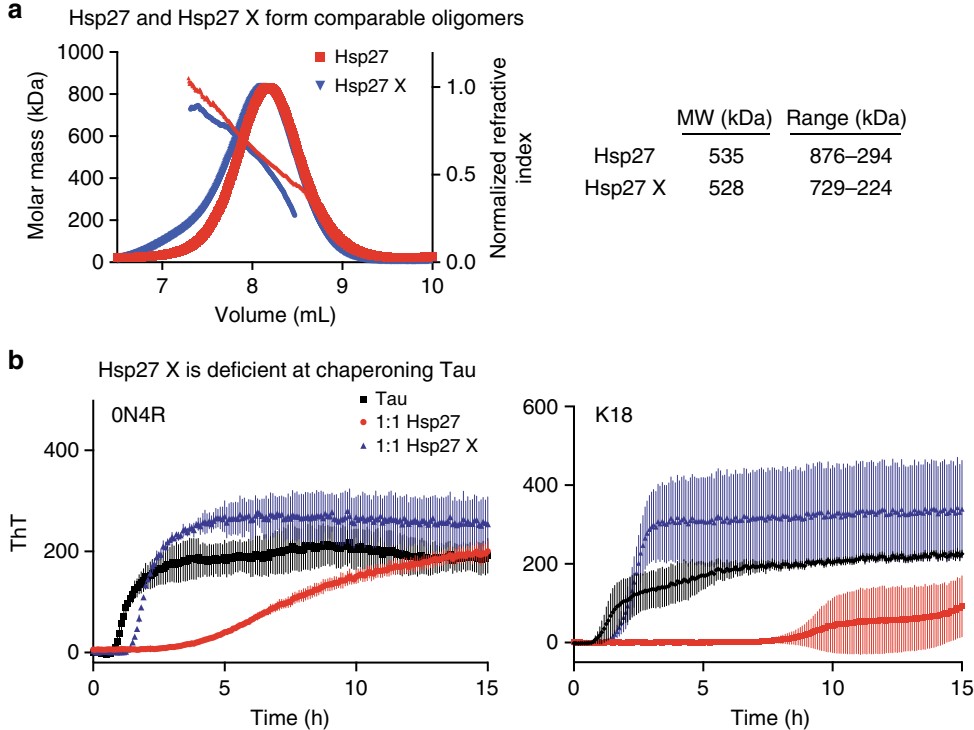

**Fig. 5** Dynamics of Hsp27 are critical to limit tau aggregation. **a** SEC-MALS trace of Hsp27 and Hsp27X (40 μM). **b** Representative aggregation curves of 0N4R (left) or K18 (right) Tau in the presence of 1:1 molar ratio of Hsp27 and Hsp27X (10 μM). Results are representative of experiments performed in at least four independent replicates. Each individual experiment was performed in triplicate and the error bars represent SEM

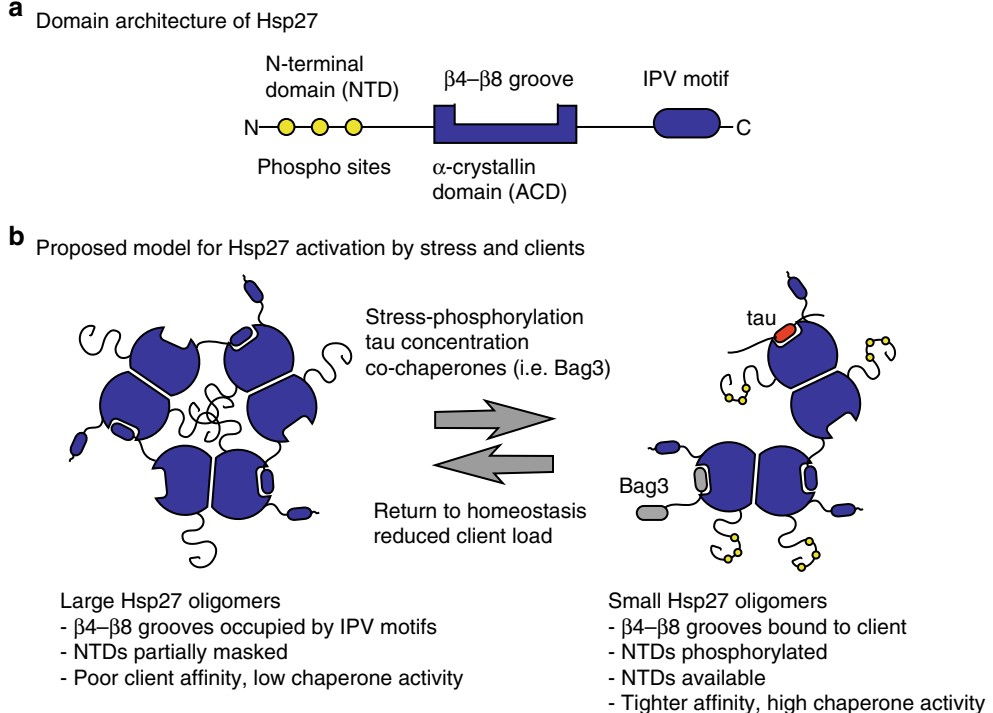

**Fig. 6** Schematic model for the activation of Hsp27. **a** Major domains of Hsp27, highlighting the sites of protein-protein interactions (PPIs). **b** Schematic summary of the model. Large Hsp27 oligomers are held together by multiple PPIs: (i) ACD–ACD interactions, (ii) self-limiting IPV contacts between dimers, and (iii) NTD interactions. Under conditions of stress, phosphorylation of the NTD, combined with competitive interactions for the β4–β8 grooves, drives oligomer disassembly and reveals the active NTDs. The contact with tau takes place within the aggregation-prone regions of the microtubule-binding repeats (red), while Bag3 uses its own IPV motifs (gray). When conditions improve, the restored self-IPV contacts would reset the system and release the client (i.e., tau). In this model, we do not want to imply any specific oligomerization state; rather, the simple trimer is used to represent what might happen in the full scope of oligomer species

Based on these results and the pioneering work of others[19,20,22,24,44], we propose a parsimonious model for Hsp27 function (Fig. 6): in its resting state Hsp27's β4/β8 groove is occupied by its own IPV motifs and it may be further auto-inhibited by additional oligomeric interactions with the NTDs. We speculate that these self-limiting PPIs are important to preclude client binding that might be detrimental to folding under normal, healthy conditions. However, under conditions of stress, the presence of higher affinity co-chaperones (e.g., BAG3)[36] and/or a large excess of clients (e.g., aggregating tau) is proposed to compete for even these multivalent PPIs. The activation of Hsp27 is likely further amplified by stress-responsive phosphorylation of the NTD, which further de-oligomerizes the complex and helps to reveal both NTDs and some β4–β8 grooves (see Fig. 4). The result of these shifting PPI priorities is that client binding in now favored. At the end of the cycle, we speculate that this same competing set of interactions might also serve to facilitate Hsp27 release from clients. In other words, as the levels of stress-responsive phosphorylation diminish and the levels of damaged client drop, the internal IPV–ACD contacts might help restore Hsp27 to its storage state. The reversibility of this regulatory step might be particularly important for sHSPs, because they lack the ATPase activity that other chaperones, such as Hsp70, use to promote cycles of client release. In this way, the sHSPs might have more in common with non-enzymatic chaperones such as trigger factor[45] and Spy[46], which use dynamic, competitive PPIs to generate reversibility.

There is great interest in either activating Hsp27 for the treatment of tauopathies[31] or inhibiting Hsp27 for the potential treatment of cancer[47] or cystic fibrosis[48]. Thus, it seems worth evaluating the implications of this model for translational research. This is especially true because Hsp27's lack of enzymatic activity implies that its PPIs may be the best sites for chemical perturbation[49]. Our findings here suggest that both the NTD and ACD contain potential sites for drug discovery.

We found that the β4/β8 groove of Hsp27 is an unexpected nexus for PPIs; a number of partners, including itself, clients, and co-chaperones (e.g., BAG3) bind the same region. While the number of binding partners for this site highlights its functional importance, it also creates a challenge for targeting the β4–β8 groove with small molecules. For example, compounds that bind this region might be counter-productive in some neurodegenerative disease settings because they would be expected to compete for the client themselves. At the same time, such compounds might partially release the self-limiting IPV–ACD contacts and, potentially, reveal the NTDs. Thus, we anticipate that molecules targeting the β4–β8 groove could be either activators or inhibitors of Hsp27 activity, depending on occupancy of the site and concentration/identity of binding partners.

Alternatively, the NTD could be a potential site for chemical intervention. This region seemed to be essential in chaperone function, at least for tau, while also playing a role in Hsp27 oligomerization. However, due to the lack of structural knowledge about this disordered region, it will likely be difficult to find a molecule that selectively interacts with it. Rather, manipulation of the NTD might be best achieved through controlling the stress pathways that tune posttranslational modifications.

Due to the conceptual difficulty in targeting either the NTD or β4–β8 groove directly, we propose that targeting the dynamics of Hsp27 may be a more fruitful approach. For example, one might stabilize specific PPIs to trap the chaperone in a specific state, favoring either client retention or release. In neurodegeneration or cancer, one might want to trap distinct states. For example, our work suggests that trapping the tau-bound Hsp27 complex might be a good way to limit tau aggregation. Thus, more broadly understanding how natural clients, in addition to tau, bind to Hsp27 and other sHSPs will likely be an important next step.

## Methods

**Cloning and recombinant protein purification.** Vectors for the expression of full-length human Hsp27 and Hsp27 ACD (residues Q80–S176) were a gift from the Klevit laboratory (University of Washington), and proteins were expressed and purified as described[50]. Hsp27 ACD L157A was generated using standard site-directed mutagenesis protocols (Supplementary Table 1). The vector for expression of Hsp27 3D was a gift from the Dickey laboratory (University of South Florida) and the protein was expressed and purified as described[31]. The Hsp27 NTD truncation, encompassing residues 1–88, was subcloned into a CMX3 vector using XhoI and Nde1 restriction enzymes. For all other Hsp27 constructs (GPG, ΔNTD consisting of residues 80–205), coding sequences were optimized for *Escherichia coli*, purchased from Integrated DNA Technologies, and LIC cloned into a PMCSG7 vector. All constructs were transformed into *E. coli* BL21 (DE3) cells, grown to an OD$_{600}$ of 0.6 in Terrific Broth (TB), and induced with 0.5 mM IPTG overnight at 18 °C. Cells were pelleted and re-suspended in 50 mM Tris-HCl pH 8.0, 10 mM imidazole, 500 mM NaCl and lysed by sonication. The cleared lysate was applied Ni-NTA resin (Qiagen) and eluted with buffer containing 300 mM imidazole. Proteins were concentrated and further purified by size exclusion on a Superdex 200 16/600 column (GE Healthcare) equilibrated with SEC buffer (50 mM sodium phosphate pH 7.5, 100 mM NaCl). Tau constructs, human 0N4R and unlabeled and $^{15}$N-labeled K18 were purified from *E. coli* BL21 (DE3)[34]. Briefly, sodium chloride (500 mM) and betaine (10 mM), were included in the TB prior to induction. Expression was induced with 200 µM IPTG for 3.5 h at 30 °C. Cells were lysed via a microfluidizer (Microfluidics) followed by boiling of lysate for 20 min. The clarified supernatant was then dialyzed overnight (buffer A: 20 mM MES pH 6.8, 50 mM NaCl, 1 mM EGTA, 1 mM MgCl, 2 mM DTT, 0.1 mM PMSF) and purified by cation exchange with a pre-elution step of 15% elution buffer (4 column volumes of buffer A with 1 M NaCl). Following a 15–60% gradient elution, pure tau fractions were pooled and concentrated prior to snap freezing aliquots for storage at −80 °C. For NMR experiments, tau was purified as above except that cells were grown in M9 minimal media containing $^{15}$NH$_4$Cl (1 g/L) and 1% glycerol. Purified $^{15}$N-labeled tau was dialyzed overnight into 20 mM ammonium bicarbonate solution then lyophilized and stored at 4 °C. All peptides were purchased from Genescript.

**NMR spectroscopy.** For Hsp27 ACD HSQC NMR experiments, all proteins were dialyzed into SEC buffer (50 mM sodium phosphate, 100 mM NaCl, pH 7.4). Then, samples containing 150 µM ACD and either 0N4R or K18 tau were prepared. Two-dimensional (2D) $^1$H–$^{15}$N hetero-nuclear single quantum coherence (HSQC) spectra were required at 32 °C on a Bruker Avance III 500 MHz spectrometer equipped with a triple resonance, z-axis gradient probe. 256 scans were acquired per t1 value for a total of 200 t1s. Spectral widths of 6009 Hz and 1419 Hz were used in the $^1$H and $^{15}$N dimensions, respectively. Spectra were processed with NMRPipe and analyzed in Sparky. Chemical shift perturbations were calculated using the following equation,

$$CSP = \sqrt{(\Delta H)^2 + (\Delta N/5)^2} \tag{1}$$

For $^{15}$N K18 HSQC NMR experiments, all proteins were dialyzed overnight into NMR buffer (25 mM HEPES pH 7.4, 10 mM KCl, 5 mM MgCl$_2$, 1 mM TCEP, 10% (v/v) D$_2$O). Then, samples containing 50 µM $^{15}$N-labeled tau and 50 or 100 µM unlabeled chaperone in NMR buffer were prepared. 2D HSQC spectra were acquired at 10 °C on a Bruker Avance AV800 spectrometer equipped with a cryoprobe. 16 scans were acquired per t1 value and spectral widths of 2100 Hz and 10,416 Hz were used in the $^1$H and $^{15}$N dimensions, respectively. Spectra were processed using rNMR and Sparky based on deposited tau assignments (Barre 2013, BioMagResBank accession number 19253). Signal intensity ratios were calculated using Prism, by dividing the intensity of chaperone-bound peaks by intensity of the same peak in the unbound spectra.

The $^{15}$N TROSY HSQC experiments were performed on a Bruker Avance 800 MHz spectrometer equipped with the TXO cryogenic probe at 283 K. The [U-$^{15}$N] K18 proteins were dissolved at a concentration of 0.05–0.15 mM in 25 mM Hepes, 10 mM KCl, 5 mM MgCl$_2$, 1 mM TCEP at pH 7.0 and titrated with the Hsp27 proteins in the same buffer. The K18:Hsp27 molar ratios in the samples are indicated in the figure legends. The $^1$H–$^{15}$N TROSY HSQC experiments were performed with 2048 direct ($^1$H) and 512 indirect ($^{15}$N) points with the spectral width of 16 ppm and 36 ppm, respectively. All NMR spectra were processed by Topspin 3.1, and analyzed with Sparky (Goddard and Kneller, SPARKY 3-NMR Assignment and Integration Software. University of California, San Francisco, CA).

**Tau aggregation assay.** Aggregation assays were performed as previously described[34]. Briefly, all proteins were dialyzed overnight at 4 °C into assay buffer (Dulbecco's PBS pH7.4, 2 mM MgCl$_2$, 1 mM DTT). 0N4R or K18 Tau (10 µM) was pre-incubated in the presence/absence of chaperones (5–20 µM) for 30 min at 37 °C. Thioflavin T (Sigma) at a final concentration of 10 µM was added and

aggregation was induced by the addition of a freshly prepared heparin sodium salt solution (Santa Cruz) at a final concentration of 44 µg/mL. For non-induced controls, assay buffer was added in place of heparin solution. The aggregation reaction was carried out at 37 °C with continuous shaking and monitored via Thioflavin T fluorescence (excitation = 444 nm, emission = 485 nm, cutoff = 480 nm) in a Spectramax M5 microplate reader (Molecular Devices). Readings were taken every 5 min for a minimum of 24 h. Baseline curves of non-induced controls (three replicates) were subtracted from induced samples (three replicates).

**Electron microscopy**. Samples of Hsp27 (1 µM, in SEC buffer) were negatively stained with uranyl formate (pH ~6.0) on thin-carbon layered 400-mesh copper grids (Ted Pella) that were glow discharged before sample was applied. Samples were imaged using a Tecnai T12 Spirit TEM (FEI) operated at 120 keV. Micrograph images were acquired with ~1.5-µm defocus on a 4k × 4k CCD camera (Gatan) at a magnification of 67,000× with a pixel size of 1.73 Å.

**SEC-MALS**. Solutions of Hsp27 were resolved by analytical SEC on a Shodex 804 column on an Ettan LC (GE Healthcare). Molecular weights were determined by multi-angle laser light scattering using an in-line DAWN HELEOS detector and an Optilab rEX differential refractive index detector (Wyatt Technology Corporation). The column was equilibrated overnight in SEC buffer. Samples were analyzed at the indicated concentrations. Calculation of molecular weights was performed using the ASTRA software package (Wyatt Technology Corporation).

**Isothermal titration calorimetry**. For experiments with BAG3, proteins were dialyzed overnight against ITC buffer (25 mM Hepes, 5 mM $MgCl_2$, 100 mM KCl, pH 7.5). For experiments with peptides, sHSPs were dialyzed overnight against SEC buffer, and peptides were dissolved in dialysis buffer. Concentrations were determined using BCA assays (Thermo Scientific), and the ITC experiment was performed with a MicroCal ITC200 (GE Healthcare) at 25 °C. In BAG3 experiments, sHSP (200 µM) in the syringe was titrated into a 10 µM cell solution of BAG3 protein. For peptide binding experiments, peptide (1 mM) was titrated into a cell containing Hsp27 proteins (100 µM). Calorimetric parameters were calculated using Origin® 7.0 software and fit with a one-site binding model.

**Thermal shift assay**. Samples of Hsp27 (25 µL; 0.3 mg/mL) in DSF buffer (50 mM NaPi pH 7.4, 700 mM NaCl, 50 mM LiCl) with a 5× final concentration of SYPRO Orange (Sigma) were placed into a white 96-well plate with optically clear caps. Melt curves were acquired on a Stratagene Mx300P RT-PCR using the SYPRO filter set. Samples were heated from 25 to 95 °C in 1 °C increments for 2 min, cooled to 25 °C at each cycle, and fluorescence was measured. The melting temperature was determined using a Boltzmann fit in Prism.

**Crosslinking of Hsp27**. Samples of Hsp27 in SEC buffer (20 µM) were cross-linked in the presence of 0.04% gluteraldehyde for 20 min, followed by quenching with 2M Tris pH 7.4. Samples were dialyzed into SEC buffer post crosslinking.

## Data availability

Data supporting the findings of this manuscript are available from the corresponding author upon reasonable request.

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

## Acknowledgements

We thank Rachel Klevit and Chad Dickey for plasmids and Matthew Ravalin for biotinylated peptides. This work was supported by NIH grant NS059690 and grants from Brightfocus Foundation and the Tau Consortium.

## Author contributions

R.F., M.B., E.T., S.-A.M., O.J., and K.T. designed and performed experiments and interpreted results. D.A.A., D.R.S., K.T., and J.E.G. provided resources and equipment. J. E.G. designed experiments and interpreted results. R.F. and J.E.G. drafted the manuscript. All authors contributed to the final manuscript.

## Additional information

**Competing interests:** The authors declare no competing interests.

