## [Peer Review File · Nature Communications]

Reviewers' Comments:

Reviewer #1:

Remarks to the Author:

The ability of some molecular chaperones to suppress protein misfolding and aggregation in the cell is very important to the maintenance of cell health and possibly also resistance to misfolding diseases such as various kinds of neurodegeneration. Large chaperones work by providing a protected environment in which a misfolded protein molecule can be pried open and allowed to reattempt the folding process. Small chaperones do not have the girth to provide such a protected environment and must go about things differently. This makes the analysis of how they work quite important. In this paper Freilich et al. look at the interactions of the small chaperone Hsp27 with tau. They use nmr and calorimetry to identify binding sites on Hsp27 for regions of tau, and determine affinities. They determine Hsp27 inhibition of tau aggregation in vitro as a "surrogate" for its possible role in vivo. The problem is important and the approaches seem to have been well-conceived and for the most part well-carried out. The results are interesting in providing a picture of the binding sites that are likely involved in the otherwise still-mysterious mechanism of Hsp27 chaperone action.

The paper includes a lot of nice data from NMR, calorimetry, and EM measurements. However, I am a little disappointed in the aggregation data as gathered by the ThT fluorescence technique. One could argue that in spite of the problems with the ThT data outlined below, they nonetheless do qualitatively demonstrate inhibition of aggregation by virtue of the observed extended ThT lag times in the presence of Hsp27, and this inhibition is, after all, the main conclusion the authors wish to draw from the ThT experiments. However, in a situation of high variability, there need to have been enough independent replicates that one is confident of any conclusions drawn. It is not clear that this is the case here. In addition, since poor reproducibility is not a quality one expects to see in an aggregation reaction that is well-set up and well-managed, one wonders exactly what the fly in the ointment might be, and what that might mean for the significance and interpretation of the results. Finally, the ThT intensities of the "inhibited" reactions many times eventually rise to significantly higher values than for the non-inhibited reactions, which at face value suggests that even if Hsp27 is a kinetic inhibitor of aggregation, at the same time it also appears to be a thermodynamic facilitator of aggregation. This seems odd and rather unlikely, but those are the data. I expand on these two issues below.

Poor reproducibility: Although lack of reproducibility in aggregation reactions is sometimes ascribed to the stochastic nature of the nucleation process, this only holds true in cases where the pseudo-steady-state concentration of newly formed nuclei is such that, at the reaction volume used, it is problematic as to whether one or more productive nuclei will emerge during the course of an experiment. This is rarely if ever the situation in the laboratory setting. Normally laboratory irreproducibility is found to be due to some lack of experimental control of the process, very often in the failure to completely eliminate from the "monomer" solutions trace amounts of aggregates that can act as seeds in the early phase of the reaction. Other possible sources of variability are irregularities in the plastic wells or in the agitation that appears to be required to drive tau aggregation. Have the authors devoted much time and energy to trying to eliminate the variability? Their one allusion to irreproducibility in the paper seems to treat it as a minor annoyance, rather than a sign of trouble. If seeds are present at the outset, of course, in the reactions being monitored here, then the whole interpretation of the results would have to change. It basically would mean that this in vitro reaction is a rather poor surrogate of the in vivo process.

It would be helpful to see the "inducer-negative" control data, in which presumably ThT is monitored over an extended period in a reaction that has not seen the heparin inducer. This control is mentioned in the text but does not seem to be included among the data. It might provide a clue as to the source of the variable results.

ThT amplitudes: It is a rare amyloid formation reaction whose thermodynamic driving force is so

weak that the reaction reaches equilibrium position at a point where substantial monomer remains; for many proteins, in fact, reversibility is difficult to observe at all, meaning reactions go to completion and there is no detectable monomer at the end and immeasurably slow dissociation of the amyloid product. Yet such an incomplete reaction is the simplistic explanation for the ThT data – that is, if the apparent “yield” of aggregates goes up by a factor of 2 in the presence of the “inhibitor” Hsp27, then in its absence the yield must have been pretty low. This is possible, but really very unlikely. The simplest explanation of the odd apparent enhancement of tau aggregation by Hsp27 is that in the presence of the chaperone, the polymorphic form of the product of tau aggregation changes, leading to an aggregate with an enhanced sensitivity to ThT. It is known that different amyloid aggregates of the same protein sequence can vary in sensitivity to ThT by over an order of magnitude! The way to check for this, and to generate data that might be shown in the paper to at least partially validate the ThT results, would be to get an independent assessment of the amount of aggregates formed at different time points. This cannot be done by light scattering, since aggregate polymorphism could greatly influence scattering intensity. The progress of the aggregation reaction will have to be assessed either by directly determining the amount of tau that has aggregated (gel on the pellet, for example), or by determining the amount of monomeric tau that remains (for example by quantitative SEC, which the authors already are using to characterize their Hsp27) and thereby indirectly quantifying aggregates. By doing such a measurement at least at the end of each reaction, it should be possible to scale the ThT amplitude, for example by simply converting the intermediate time point values to “% aggregation”.

Alternative explanations for the enhanced ThT signal in the “inhibited” reactions might be that Hsp27 is co-aggregating with tau in such a way as to add additional ThT-sensitive aggregate mass; this could be checked by running a gel on a centrifugation pellet of the aggregation reaction. Yet again, Hsp27 may itself independently aggregate under these conditions to produce ThT-positive aggregates. Has that control been done?

Minor point: it would be useful if the authors provided the actual weight concentrations of tau and HSP27 used in the aggregation reactions (not just their weight or molar ratio), as well as the derived molar concentrations of tau. Since the authors have nicely determined the binding constants of tau for the sites on Hsp27, it is of some mechanistic interest to see whether one expects tau binding to be saturated, partial, or poor under the conditions of the aggregation reactions.

Reviewer #2:

Remarks to the Author:

Freilich and colleagues report on studies investigating the mechanism of interaction of the small heat shock protein Hsp27 with the microtubule-associated protein Tau. Aggregation of Tau plays an important role in Alzheimer's disease and previous studies already showed that Hsp27 delays tau amyloid formation. The current work tries to better understand which regions/domains of Hsp27 are important for delay of tau fibril formation and how this might be connected to the well-known propensity of Hsp27 to oligomerize. Through a number of binding studies and supported by ThT aggregation experiments the authors conclude that Hsp27 oligomerization/self-associations decreases binding to Tau and its effect on tau aggregation, while a decrease in Hsp27 self-association enhances tau binding. In addition, the authors find that the NTD of Hsp27 is important for chaperoning of tau, while the most specific tau interaction occurs with the ACD of Hsp27. Overall, the work is well performed and concludes with an interesting model.

Points to address:

1) The authors are not able to quantify the affinity between ACD and K18/tau and conclude that the affinity is lower than for the IPV motifs. However, comparison of Fig. 1b and 1c rather suggests that K18 has a higher affinity for ACD when compared to the IPV peptide. This inconsistency should be clarified and the affinity of K18 for Hsp27 should be quantified (e.g. by

fluorescence anisotropy).

2) The authors suggest that the interaction between K18 and the NTD is electrostatic in nature. This should be supported by additional interaction studies at increasing ionic strength.

3) Hsp27 and Hsp27 3D oligomers are compared by EM (Fig. 3B). EM is however not quantitative. SEC-MALS studies are required for Hsp27 and Hsp27 3D.

Reviewer #3:

Remarks to the Author:

The manuscript has significant flaws and caveats (i.e. some of the experimental sections of the data presented are missing or are lacking detail; some 'seemingly' biased interpretation of the data; and the use of SEM which in my opinion should be SD as most of the data shown are technical replicates).

The final figure 6 showing a model of a symmetrical trimer (yet describing it as an oligomer) is misleading and is not substantiated by the data presented in the SI nor main figures. Indeed, Why were full length hsp27 and GPG not used for direct comparison of SEC-MALS with EM data in Figure S5? Full length proteins show opposite results compared to the deletion mutants used in Panel B but these data are not really discussed in the text?

"Hsp27 GPG has an altered oligomeric structure as compared to WT (Figure S5A) and, as expected, it has a lower apparent melting temperature, as measured by differential scanning fluorimetry (DSF) (Figure 4A)".

The authors totally ignore this important finding? Why is this in the SI? This finding is in contradiction with oligomeric state of DNTD hsp27 and DNTD GPG hsp27. Please explain?

"Thus, smaller oligomers of Hsp27 seemed to have more exposed b4-b8 grooves."

Please provide experimental evidence for this claim.

The data in Figure S3 is not reflecting Figure 4C. It seems that different residue numbers in Tau interact with GPG vs WT Hsp27 despite the fact that the same residues in GPG and WT hsp27 interact with Tau.

Please provide the experimental details on the crosslinking experiments. In line, please provide more experimental details in your Electron Microscopy section regarding buffer compositions and protein concentrations.

Regarding the Bilayer Interferometry: Which data is shown and where is the control data?

Dilutions of protein in BLI Buffer is not explained? What is the source concentration and buffer composition of the protein. This can produce concentration dependent bulk shifts. What is the mode of binding and residuals, number of repeats?

The discussion should be restructured to discuss the actual importance of the results and not just merely become a highly speculative means to sell the putative merits of the findings in drug discovery.

Figure 1:

Provide a metaschematic map of all constructs used in this paper and their acronyms as well as their schematic structures used in the following figures to avoid confusing the readers. Align them to indicate all important residues and boundaries. Apply the same acronyms throughout to avoid confusing the reader.

Figure 1B: Why are the chemical shift perturbations no more than 0.1? How would another protein

in the same buffer behave? What are the contributions of the buffer itself?

Figure 1D: Please show ITC data for this.

Figure 1E: Please show the calculations and fit for this figure. In addition what was the peptide sequence used and was there a negative control?

Figure 1E:

What is the peptide that was used? What is the response with a random negative control peptide loaded at the same level? What do the authors mean with K_D estimated at 15 μM ? This data does not reflect a protein binding to a peptide. What is the loading of the peptide in response (nm)? What is the expected response of ACD (nm) at saturation based on peptide loading and difference in MW?

It could be that what is measured is increasing bulk shift due to differences in buffer with increasing concentrations. What is the expected plateau at saturation from the model

Figure 3C: where are the controls of Hsp27 and hsp27 3D?

Figure 4A: Why were the T_m of the other hsp27 variants not analysed? I can't see the relevance of these data without any IPV peptide binding data. These experiments could have supported the ITC data

Figure 5A: Why is there no indication on the Molar mass of oligomers? I assume red is hsp27 and blue is hsp27 X? Where the proteins checked for crosslinking by SDS-PAGE?

Figure 6B: There is no data presented here supporting this discrete trimer model of hsp27! This model is speculative and should be removed as it will mislead readers!

Figure S5A: Be specific about the significant increased oligomeric state of GPG. Elaborate as to why you would expect a lower apparent T_m for GPG if it forms much larger oligomers?

We thank the Editor and reviewers for the careful analysis and useful comments. Herein, we provide a point-by-point summary of the changes made to the manuscript in response to these comments. We have added substantial data to the figures and supplemental figures, which give considerable confidence in the original conclusions. Here, the reviewer comment is reproduced in italics and the response in blue.

Reviewers' comments:

Reviewer #1 (Remarks to the Author):

The ability of some molecular chaperones to suppress protein misfolding and aggregation in the cell is very important to the maintenance of cell health and possibly also resistance to misfolding diseases such as various kinds of neurodegeneration. Large chaperones work by providing a protected environment in which a misfolded protein molecule can be pried open and allowed to reattempt the folding process. Small chaperones do not have the girth to provide such a protected environment and must go about things differently. This makes the analysis of how they work quite important. In this paper Freilich et al. look at the interactions of the small chaperone Hsp27 with tau. They use nmr and calorimetry to identify binding sites on Hsp27 for regions of tau, and determine affinities. They determine Hsp27 inhibition of tau aggregation in vitro as a "surrogate" for its possible role in vivo. The problem is important and the approaches seem to have been well-conceived and for the most part well-carried out. The results are interesting in providing a picture of the binding sites that are likely involved in the otherwise still-mysterious mechanism of Hsp27 chaperone action.

The paper includes a lot of nice data from NMR, calorimetry, and EM measurements. However, I am a little disappointed in the aggregation data as gathered by the ThT fluorescence technique. One could argue that in spite of the problems with the ThT data outlined below, they nonetheless do qualitatively demonstrate inhibition of aggregation by virtue of the observed extended ThT lag times in the presence of Hsp27, and this inhibition is, after all, the main conclusion the authors wish to draw from the ThT experiments. However, in a situation of high variability, there need to have been enough independent replicates that one is confident of any conclusions drawn. It is not clear that this is the case here. In addition, since poor reproducibility is not a quality one expects to see in an aggregation reaction that is well-set up and well-managed, one wonders exactly what the fly in the ointment might be, and what that might mean for the significance and interpretation of the results. Finally, the ThT intensities of the "inhibited" reactions many times eventually rise to significantly higher values than for the non-inhibited reactions, which at face value suggests that even if Hsp27 is a kinetic inhibitor of aggregation, at the same time it also appears to be a thermodynamic facilitator of aggregation. This seems odd and rather unlikely, but those are the data. I expand on these two issues below.

Poor reproducibility: Although lack of reproducibility in aggregation reactions is sometimes ascribed to the stochastic nature of the nucleation process, this only holds true in cases where the pseudo-steady-state concentration of newly formed nuclei is such that, at the reaction volume used, it is problematic as to whether one or more productive nuclei will emerge during the course of an experiment. This is rarely if ever the situation in the laboratory setting. Normally laboratory irreproducibility is found to be due to some lack of experimental control of the process, very often in the failure to completely eliminate from the “monomer” solutions trace amounts of aggregates that can act as seeds in the early phase of the reaction. Other possible sources of variability are irregularities in the plastic wells or in the agitation that appears to be required to drive tau aggregation. Have the authors devoted much time and energy to trying to eliminate the variability? Their one allusion to irreproducibility in the paper seems to treat it as a minor annoyance, rather than a sign of trouble. If seeds are present at the outset, of course, in the reactions being monitored here, then the whole interpretation of the results would have to change. It basically would mean that this in vitro reaction is a rather poor surrogate of the in vivo process.

It would be helpful to see the “inducer-negative” control data, in which presumably ThT is monitored over an extended period in a reaction that has not seen the heparin inducer. This control is mentioned in the text but does not seem to be included among the data. It might provide a clue as to the source of the variable results.

>> We appreciate the reviewer’s careful attention to an important point and regret our wording in the original text. Briefly, we meant to refer to reproducibility issues arising from Hsp27 and not from tau. Indeed, exactly as the reviewer suggests, we have observed that tau itself displays quite reproducible aggregation behavior *in vitro* (see below; black). However, Hsp27 forms an ensemble of oligomers, introducing a slight, and expected, variation in behavior between protein batches. What we meant to conclude in the original text was that these properties prevent us from quantifying a statistically significant difference between Hsp27 and Hsp27 3D’s efficacy by ThT assays. We have updated the text to make this point more clearly.

The reviewer also makes good points about additional controls that should accompany the manuscript. In each experiment, we prepare parallel wells lacking heparin sulfate. This background is small (<10%; see below), but we subtracted the value from the remaining experiments. To the point above, one can notice the relatively higher variability from different batches of Hsp27 in these experiments.

ThT amplitudes: It is a rare amyloid formation reaction whose thermodynamic driving force is so weak that the reaction reaches equilibrium position at a point where substantial monomer remains; for many proteins, in fact, reversibility is difficult to observe at all, meaning reactions go to completion and there is no detectible monomer at the end and immeasurably slow dissociation of the amyloid product. Yet such an incomplete reaction is the simplistic explanation for

the ThT data – that is, if the apparent “yield” of aggregates goes up by a factor of 2 in the presence of the “inhibitor” Hsp27, then in its absence the yield must have been pretty low. This is possible, but really very unlikely. The simplest explanation of the odd apparent enhancement of tau aggregation by Hsp27 is that in the presence of the chaperone, the polymorphic form of the product of tau aggregation changes, leading to an aggregate with an enhanced sensitivity to ThT. It is known that different amyloid aggregates of the same protein sequence can vary in sensitivity to ThT by over an order of magnitude! The way to check for this, and to generate data that might be shown in the paper to at least partially validate the ThT results, would be to get an independent assessment of the amount of aggregates formed at different time points. This cannot be done by light scattering, since aggregate polymorphism could greatly influence scattering intensity. The progress of the aggregation reaction will have to be assessed either by directly determining the amount of tau that has aggregated (gel on the pellet, for example), or by determining the amount of monomeric tau that remains (for example by quantitative SEC, which the authors already are using to characterize their Hsp27) and thereby indirectly quantifying aggregates. By doing such a measurement at least at the end of each reaction, it should be possible to scale the ThT amplitude, for example by simply converting the intermediate time point values to “% aggregation”.

Alternative explanations for the enhanced ThT signal in the “inhibited” reactions might be that Hsp27 is co-aggregating with tau in such a way as to add additional ThT-sensitive aggregate mass; this could be checked by running a gel on a centrifugation pellet of the aggregation reaction. Yet again, Hsp27 may itself independently aggregate under these conditions to produce ThT-positive aggregates. Has that control been done?

>> Again, we agree with the reviewer and apologize for not making our interpretation more clear. Indeed, we know that a portion of the Hsp27 molecules assemble into tau fibrils, as judged by their co-centrifugation (see below; S = soluble and P = pellet). This result was predicted by the reviewer and it is the most likely contributor to the apparent increase in ThT signal. While we cannot rule out that the possibility that Hsp27 effects the polymorphic form of tau aggregates, we think that incorporation of Hsp27 is the likely culprit. We have updated the text to make these points more clear.

Minor point: it would be useful if the authors provided the actual weight concentrations of tau and HSP27 used in the aggregation reactions (not just their weight or molar ratio), as well as the derived molar concentrations of tau. Since the authors have nicely determined the binding constants of tau for the sites on Hsp27, it is of some mechanistic interest to see whether one expects tau binding to be saturated, partial, or poor under the conditions of the aggregation reactions.

>> Good point. We have changed the figure legends to show both concentrations and molar ratios. Related to the other point, it is unsurprising that the affinity of the interactions did not seem to be the only contributor to anti-aggregation potency. For example, while the ACD has the strongest interaction affinity for tau, it is unable to block aggregation (because it lacks NTDs). In other words, binding affinity is certainly important, but the correct positioning of multiple domains is also critical.

Reviewer #2 (Remarks to the Author):

Freilich and colleagues report on studies investigating the mechanism of interaction of the small heat shock protein Hsp27 with the microtubule-associated protein Tau. Aggregation of Tau plays an important role in Alzheimer's disease and previous studies already showed that Hsp27 delays tau amyloid formation. The current work tries to better understand which regions/domains of Hsp27 are

important for delay of tau fibril formation and how this might be connected to the well-known propensity of Hsp27 to oligomerize. Through a number of binding studies and supported by ThT aggregation experiments the authors conclude that Hsp27 oligomerization/self-associations decreases binding to Tau and its effect on tau aggregation, while a decrease in Hsp27 self-association enhances tau binding. In addition, the authors find that the NTD of Hsp27 is important for chaperoning of tau, while the most specific tau interaction occurs with the ACD of Hsp27. Overall, the work is well performed and concludes with an interesting model.

Points to address:

1) *The authors are not able to quantify the affinity between ACD and K18/tau and conclude that the affinity is lower than for the IPV motifs. However, comparison of Fig. 1b and 1c rather suggests that K18 has a higher affinity for ACD when compared to the IPV peptide. This inconsistency should be clarified and the affinity of K18 for Hsp27 should be quantified (e.g. by fluorescence anisotropy).*

>> The NMR data in Fig 1b and 1c is performed at single concentrations, so does not directly inform on relative affinity. Rather, we attempted the more quantitative assays of ELISA competition (below, left) and ITC (below, right). Our findings in those platforms support the idea that Hsp27 ACD binds, but that the affinity is weak ($>100 \mu\text{M}$). This finding is also in agreement with literature precedent, where anisotropy titration of the Hsp27-tau interaction was not saturable (reference 33).

2) *The authors suggest that the interaction between K18 and the NTD is electrostatic in nature. This should be supported by additional interaction studies at increasing ionic strength.*

>> This is a good idea. We tried to perform the ^{15}N K18 NMR and ThT experiments at higher salt concentration, but found that the salt interfered with the signal in both cases. Accordingly, we have removed this suggestion from the text. Fortunately, this point was speculative and not related to the main conclusions.

3) Hsp27 and Hsp27 3D oligomers are compared by EM (Fig. 3B). EM is however not quantitative. SEC-MALS studies are required for Hsp27 and Hsp27 3D.

>> We have included the requested experiment in the new figure 3B. These results support the idea that Hsp27 3D is more heterogeneous than wild type, forming both larger and smaller isoforms. These experiments clarify the complex roles of Hsp27 3D in the system and we appreciate the reviewer's request.

Reviewer #3 (Remarks to the Author):

The manuscript has significant flaws and caveats (i.e. some of the experimental sections of the data presented are missing or are lacking detail; some 'seemingly' biased interpretation of the data; and the use of SEM which in my opinion should be SD as most of the data shown are technical replicates).

The final figure 6 showing a model of a symmetrical trimer (yet describing it as an oligomer) is misleading and is not substantiated by the data presented in the SI nor main figures.

>> We apologize for the apparent confusion. We did not intend for this model to imply a distinct oligomeric state (e.g. trimer, tetramer, etc), but rather to represent what might be happening in the full range of oligomer types in the simplest possible form. This approach to model building is common in the small heat shock protein field, due to the heterogeneity of architectures. We have adjusted the Figure 6 legend to indicate this point explicitly.

Indeed, Why were full length hsp27 and GPG not used for direct comparison of SEC-MALS with EM data in Figure S5? Full length proteins show opposite results compared to the deletion mutants used in Panel B but these data are not really discussed in the text?

>> Unfortunately, we could not compare Hsp27 GPG to other constructs using SEC-MALS because it is too large (it appears in the void volume). That being said, we still find that the current results provide significant insight. Specifically, the DSF shows that Hsp27 GPG is less stable (lower melting temperature) and the EM shows that average oligomers are apparently larger in size. One way to explain both of these findings is that the lack of an IPV motif creates oligomers with approximately similar stoichiometry but less packing/density. This model is consistent with the known role of the IPV in being one of the protein-protein interactions that stabilize inter-protomer contacts.

*"Hsp27 GPG has an altered oligomeric structure as compared to WT (Figure S5A) and, as expected, it has a lower apparent melting temperature, as measured by differential scanning fluorimetry (DSF) (Figure 4A)".
The authors totally ignore this important finding? Why is this in the SI? This*

finding is in contradiction with oligomeric state of DNTD hsp27 and DNTD GPG hsp27. Please explain?

>> As mentioned above, we think that deleting the IPV motif damages one of the multiple inter- and intra-protomer contacts in Hsp27 – resulting in an oligomer with lower stability, but that is less densely packed. This possibility has been included in the revised text, although it will require many additional experiments to confirm this idea.

“Thus, smaller oligomers of Hsp27 seemed to have more exposed b4-b8 grooves.” Please provide experimental evidence for this claim.

>> This statement refers to the set of experiments in which binding of IPV peptides to Hsp27 variants is measured by ITC. The results show that smaller oligomers (i.e., ACD, Δ NTD WT and Δ NTD GPG) have a relatively tighter affinity than larger oligomers (WT). We have clarified this statement in the text to read: “Thus, it seems possible that more β 4- β 8 grooves may be available for binding in the absence of NTDs”.

The data in Figure S3 is not reflecting Figure 4C. It seems that different residue numbers in Tau interact with GPG vs WT Hsp27 despite the fact that the same residues in GPG and WT hsp27 interact with Tau.

>> Figure 4C shows that Hsp27 GPG binds to the VQIINK and VQIVYK motifs, while Hsp27 WT does not. This conclusion is also supported by Figure S3, in which Hsp27 GPG binds, while Hsp27 WT does not. In these two experiments, we did note that Hsp27 GPG had additional contacts outside the aggregation motifs (for example, in the start of R2 and R4). These “extra” contacts are more prominent in Figure 4C than in Figure 3C, likely because of differences in the sensitivity of the platforms. However, this difference has no effect on any of our conclusions.

Please provide the experimental details on the crosslinking experiments.

>> We have included a new section in the revised Methods section.

In line, please provide more experimental details in your Electron Microscopy section regarding buffer compositions and protein concentrations.

>> Additional details regarding the buffer/concentration have been added to the revised Methods.

Regarding the Bilayer Interferometry: Which data is shown and where is the control data? Dilutions of protein in BLI Buffer is not explained? What is the source concentration and buffer composition of the protein. This can produce concentration dependent bulk shifts. What is the mode of binding and residuals,

number of repeats?

>> We have removed the BLI data from the manuscript, as it was tangential to the points of the manuscript.

The discussion should be restructured to discuss the actual importance of the results and not just merely become a highly speculative means to sell the putative merits of the findings in drug discovery.

>> This project was initiated and carried out with the primary purpose of understanding whether the Hsp27-tau complex might serve as a drug target. This focus arises from the fact that our laboratory is a chemical biology group, which has produced multiple clinical-stage assets. Thus, we are comfortable with the focus of the discussion.

Figure 1:

Provide a metaschematic map of all constructs used in this paper and their acronyms as well as their schematic structures used in the following figures to avoid confusing the readers. Align them to indicate all important residues and boundaries. Apply the same acronyms throughout to avoid confusing the reader.

>> This is a good idea. We have added a schematic of all constructs in revised figure S1.

Figure 1B: Why are the chemical shift perturbations no more than 0.1? How would another protein in the same buffer behave?

>> The observed CSPs are in-line with expectations, based on published reports (Mok et al 2018 Nat Struct. Mol. Biol). We have added an additional experiment

in which Hsp27 ACD binding to a poly-glutamine expanded Huntingtin exon 1 (both 25Q and 46Q) was measured as a negative control. Consistent with the model, no significant CSPs or intensity changes were observed (see figure).

What are the contributions of the buffer itself?

>> All proteins were dialyzed into the same batch of buffer (50 mM NaPi, 100 mM NaCl, pH7.4) prior to NMR experiments to minimize buffer effects (this is reflected in the revised Methods). For every NMR experiment, we compare spectra to the buffer only control.

Figure 1D: Please show ITC data for this.

>> We have now included a representative curve for each peptide in the revised Figure S2.

Figure 1E: Please show the calculations and fit for this figure. In addition what

was the peptide sequence used and was there a negative control?

Figure 1E: What is the peptide that was used? What is the response with a random negative control peptide loaded at the same level? What do the authors mean with KD estimated at 15 μ M? This data does not reflect a protein binding to a peptide. What is the loading of the peptide in response (nm)? What is the expected response of ACD (nm) at saturation based on peptide loading and difference in MW? It could be that what is measured is increasing bulk shift due to differences in buffer with increasing concentrations. What is the expected plateau at saturation from the model

>> As mentioned above, we have removed the BLI from the figure, as it was tangential to the conclusions.

Figure 3C: where are the controls of Hsp27 and hsp27 3D?

>> In Figure 3C, the Hsp27 serves as the control for Hsp27 3D. As mentioned above, each spectra was compared to a buffer control.

Figure 4A: Why were the Tm of the other hsp27 variants not analysed? I can't see the relevance of these data without any IPV peptide binding data. These experiments could have supported the ITC data

>> The original goal of Figure 4A was simply to ask whether Hsp27 GPG was more or less stable than Hsp27 WT. The reviewer seems to want us to use this platform to support the ITC in Figure 1D or the NMR studies in Figure 1C. Accordingly, we have included the DSF-derived melting curves for Hsp27 3D in the revised Figure 3. Unfortunately, other sHSPs gave multiple transitions, making it more difficult to interpret the results. Moreover, we did attempt to measure IPV peptide binding in this platform, but had difficulty interpreting data as the shape of curves was significantly affected by peptide (see below).

Figure 5A: Why is there no indication on the Molar mass of oligomers? I assume red is hsp27 and blue is hsp27 X? Where the proteins checked for crosslinking by SDS-PAGE?

>> We apologize for the lack of labels. We have now included the calculated, average molar mass of oligomers and the identity of the curves. Crosslinking was measured by SDS-PAGE and is now shown in figure S7.

Figure 6B: There is no data presented here supporting this discrete trimer model of hsp27! This model is speculative and should be removed as it will mislead readers!

>> See comments above.

Figure S5A: Be specific about the significant increased oligomeric state of GPG. Elaborate as to why you would expect a lower apparent T_m for GPG if it forms much larger oligomers?

>> See comments above.

Reviewers' Comments:

Reviewer #1:

Remarks to the Author:

I appreciate the comments of the authors to my concerns and in most ways I find them satisfying. However, one thing that concerns me is that, in my view, it is not rigorous to supply such comments and additional data only to the reviewer. For example, I believe that any worker in the protein aggregation field reading this paper in its present form would be struck by the much higher amplitude seen in the ThT curves for the "+ chaperone" samples, and would have the same legitimate concerns as I expressed in my review. It seems to me that private assurances to the reviewer are inadequate. And even if my objections, and the authors' rebuttals, become part of the published record, in many cases it is likely that readers will only download and focus on the paper itself. The actual paper remains the principle vehicle for the authors data and ideas, and should contain any important caveats.

For example, in the case of the ThT amplitudes, the authors need to revise their paper by including the following: (a) they need to acknowledge the inexplicably higher amplitude in the +chaperone curves, and discuss the two generic possibilities of (1) co-aggregation of the chaperone or (2) formation of a higher-ThT yield polymorphic tau fibril. Then (2) they need to show the gel result they provided me, perhaps in additional supplemental data; by the way, this gel is not really very well annotated or explained by the authors in their rebuttal, and this should be fixed, even if only in the published rebuttal; right now it is totally ambiguous as to what the gel shows. For example, what is the difference between the two gels that are shown side-by-side? As part of this exercise, they should do densitometry to get an approximate weight and molar ratio of chaperone that is now revealed for the first time to be co-aggregating with the tau. This is an important result! It should not be swept under the rug!!

Similarly, the fact that the concentrations used in the aggregation experiments are not expected to give much binding based on the deduced binding constants has to be reported and discussed in the main text discussion. It is important information that must somehow be rationalized in a satisfactory way. The authors can certainly then provide their explanation for how this doesn't necessarily mean that protective binding is not technically possible, and the reader can come to their own judgment. (Personally I find their rationale unconvincing: it is fine to say that orientation is as important as binding, but orientation only can play a role IF THERE IS BINDING, and the disconnect between binding constants and the concentrations of the aggregation experiments remain.) This is another important result that readers need to be aware of when they try to gauge the significance of the work and interpretations.

Reviewer #2:

Remarks to the Author:

I am a bit disappointed by the authors' reply to my comments, because they chose to only address one of my three comments:

1) The authors argue that they tried to use ELISA and ITC (which did not work) to support their statement. However, the NMR data are still in disagreement with their conclusion. If (as the authors say) they only measured the NMR data at one concentration and they believe this is not trust-worthy/reliable, then they should do a full titration. If they don't feel confident to do additional NMR measurements, there are other techniques such as SPR, thermofluor, ... to quantify affinities.

2) Higher salt: It is kind of surprising that the authors first make a claim about ionic strength dependence, but don't want to support it by a simple salt titration. This should be performed for the revised version of the manuscript.

We sincerely apologize to the Reviewers and Editor for errors made in the previous revision, which caused unnecessary confusion. We thank you for the patience. Below, we included a point-by-point summary of the most recent changes made in blue.

Reviewer #1 (Remarks to the Author):

I appreciate the comments of the authors to my concerns and in most ways I find them satisfying. However, one thing that concerns me is that, in my view, it is not rigorous to supply such comments and additional data only to the reviewer. For example, I believe that any worker in the protein aggregation field reading this paper in its present form would be struck by the much higher amplitude seen in the ThT curves for the "+ chaperone" samples, and would have the same legitimate concerns as I expressed in my review. It seems to me that private assurances to the reviewer are inadequate. And even if my objections, and the authors' rebuttals, become part of the published record, in many cases it is likely that readers will only download and focus on the paper itself. The actual paper remains the principle vehicle for the authors data and ideas, and should contain any important caveats.

For example, in the case of the ThT amplitudes, the authors need to revise their paper by including the following: (a) they need to acknowledge the inexplicably higher amplitude in the +chaperone curves, and discuss the two generic possibilities of (1) co-aggregation of the chaperone or (2) formation of a higher-ThT yield polymorphic tau fibril. Then (2) they need to show the gel result they provided me, perhaps in additional supplemental data; by the way, this gel is not really very well annotated or explained by the authors in their rebuttal, and this should be fixed, even if only in the published rebuttal; right now it is totally ambiguous as to what the gel shows. For example, what is the difference between the two gels that are shown side-by-side? As part of this exercise, they should do densitometry to get an approximate weight and molar ratio of chaperone that is now revealed for the first time to be co-aggregating with the tau. This is an important result! It should not be swept under the rug!!

>> No problem! We are happy to include a summary of the previous discussion in the revised manuscript. Frankly, it simply didn't occur to us. In the revised manuscript, we have now added text (page 8) on why there might be a higher amplitude of the ThT signal in some experiments. We have also added the data on incorporation of Hsp27 into insoluble tau fibrils (with better labels) into Figure S7C. Finally, we adjusted the legend of Figure 3 and Figure S7 to acknowledge the higher amplitudes and provide generic possibilities for why they might occur. We thank the reviewer for the idea to expand this part of the work.

Similarly, the fact that the concentrations used in the aggregation experiments are not expected to give much binding based on the deduced binding constants has to be reported and discussed in the main text discussion. It is important information that must somehow be rationalized in a satisfactory way. The authors can certainly then provide their explanation for how this doesn't necessarily mean that protective binding is not technically possible, and the reader can come to their own judgment. (Personally I find their rationale unconvincing: it is fine to say that orientation is as important as binding, but orientation only can play a role IF THERE IS BINDING, and the disconnect between binding constants and the concentrations of the aggregation experiments remain.) This is another important result that readers need to be

aware of when they try to gauge the significance of the work and interpretations.

>> We have taken this comment (and related comments of Reviewer 2; below) seriously. With the goal of making the interpretations clearer, we have significantly altered the text. Clearly, (as the Reviewer says) binding of Hsp27 to tau is absolutely required for anti-aggregation activity. We did not intend to suggest otherwise. Rather, the parsimonious model we propose is based on the observations that the individual PPIs in this system (*e.g.* a single IPV peptide binding to Hsp27 ACD or monomeric tau binding to Hsp27 ACD) are weak ($> 100 \mu\text{M}$). However, the multivalent interactions, such as those involving multiple IPV motifs within a large Hsp27 oligomer, seem to be strong. In this way, we think that the self-interactions within Hsp27 keep it “off” until the system is confronted with aggregating tau (which now presents its own multivalent surface too). This is an elegant way for a non-enzymatic chaperone to be regulated and it is reminiscent of what has been proposed for Trigger Factor and Spy. We have made changes throughout the Results and Discussion to better reflect this important idea (see also below and Tracked Changes in the Word document).

Reviewer #2 (Remarks to the Author):

I am a bit disappointed by the authors' reply to my comments, because they chose to only address one of my three comments:

1) The authors argue that they tried to use ELISA and ITC (which did not work) to support their statement. However, the NMR data are still in disagreement with their conclusion. If (as the authors say) they only measured the NMR data at one concentration and they believe this is not trust-worthy/reliable, then they should do a full titration. If they don't feel confident to do additional NMR measurements, there are other techniques such as SPR, thermofluor, ... to quantify affinities.

>> In looking back at our previous response to the Reviewer's original comments, we deeply regret not providing additional, clarifying information. In addition, we noticed a serious error in the submitted text that generated unnecessary confusion. In this latest revision, we have carefully changed the wording (see bullet points below) to remove and clarify this apparent contradiction.

Briefly, the major source of confusion seems to be that we incorrectly stated that tau binds weaker to Hsp27's ACD than the IPV peptide. As the Reviewer says, this is not true. Indeed, previous work had already concluded that binding of monomeric tau to Hsp27 ACD is also weak (Baughman et al. 2018 J. Biol. Chem.). Moreover, our own data supports that conclusion (see below). Specifically, we find that both tau ($K_d > 100 \mu\text{M}$; Fig S3) and Hsp27's IPV bind weakly ($K_d > 25 \mu\text{M}$; Fig 1D) to Hsp27's ACD. These affinities are both weak and not readily discernable. We have fixed the errors in the revised text and sincerely apologize.

To further address the Reviewer's point, we have also carried out additional experiments and added them to the new Figure S3. Briefly, these include (a) ELISA studies to show that Hsp27 ACD has weak affinity for tau ($\text{IC}_{50} > 100 \mu\text{M}$), (b) ITC studies to show that K18 tau binding to Hsp27's ACD is not within the range of that method ($K_d > 100 \mu\text{M}$), (c) NMR titrations of Hsp27 ACD into ^1H - ^{15}N K18 tau to estimation a weak (*i.e.* unsaturable) affinity, based on the CSPs in Val63 and (d) titration of tau-derived VQIVYK peptide into ^1H - ^{15}N Hsp27 ACD, showing weak/no binding. Together, these results more firmly support the conclusion that the individual PPIs in

this system are weak in isolation. As discussed above and clarified in the revised text, they seem to become much stronger in the context of multivalent contacts.

Finally, we have adjusted the text to make more explicitly clear when the His-containing IPV peptide (EITIPVTHE; a chimera with BAG3's tighter binding IPV) is being used as a chemical probe instead of the native Hsp27 sequence (EITIPVTFE) (page 10). As shown in Figure 1, this chimera has a tighter affinity than the native peptide.

2) Higher salt: It is kind of surprising that the authors first make a claim about ionic strength dependence, but don't want to support it by a simple salt titration. This should be performed for the revised version of the manuscript.

>> We apologize for the brevity of our previous response to this request. In retrospect, we did not properly describe our complete logic. As a reminder, in the first submission, we had briefly (in a sentence fragment) speculated that, perhaps, the N-terminal domain (NTD) might use electrostatic interactions to bind tau. This conjecture was not based on any data; rather, it was just an extrapolation of the fact that both NTD and tau are known to be charged. In the first round of review, the reviewer rightfully asked us to justify this speculation. At the time, we regretted saying anything at all about the unfounded possibility of electrostatics and chose to simply remove the statement. However, the decision to remove the mention of electrostatics, rather than experimentally address it, was based on more technical knowledge than this. We regret not expanding on the logic.

The fact is that a "simple" salt experiment would not work for this system because the ACD-ACD contact is also held together by salt-bridges. In other words, as salt concentration is changed, it would impact multiple PPIs (e.g. ACD-ACD, NTD-tau, etc) simultaneously and make any results un-interpretable. Rather, in more recent discussions with our team, we envisioned that the best (perhaps only) way to address the role of electrostatics would be to systematically mutate each of the charged residues in Hsp27's NTD (and/or tau). While this is a potentially interesting set of experiments, it seems to go beyond the scope of the current manuscript. We sincerely hope that the Reviewer finds this more detailed description of why we cannot quickly interrogate the role of electrostatics satisfactory. Again, we are sorry for seeming to ignore the suggestion, which was not intended.

Reviewers' Comments:

Reviewer #1:

Remarks to the Author:

This reviewer appreciates the new discussion points added to the manuscript itself, and suspects that at least some readers will also appreciate the new discussion. I'm happy the authors included the key phrase "avidity effect" which really captures, in a familiar biochemical term, the essence of their argument on how a weak binding interaction can still be observed at low protein concentrations. My only suggestion for a further small change is that, in the new footnote to Figure S7, they emphasize that they are speculating that the reason for the higher ThT signal is due to the co-aggregation of chaperone "in an amyloid-like, ThT-positive manner" - or words to that effect.

Reviewer #2:

Remarks to the Author:

Thanks for the clarification. The authors have fully addressed my concerns.

REVIEWERS' COMMENTS:

Reviewer #1 (Remarks to the Author):

This reviewer appreciates the new discussion points added to the manuscript itself, and suspects that at least some readers will also appreciate the new discussion. I'm happy the authors included the key phrase "avidity effect" which really captures, in a familiar biochemical term, the essence of their argument on how a weak binding interaction can still be observed at low protein concentrations. My only suggestion for a further small change is that, in the new footnote to Figure S7, they emphasize that they are speculating that the reason for the higher ThT signal is due to the co-aggregation of chaperone "in an amyloid-like, ThT-positive manner" - or words to that effect. >> Again, we thank the reviewer for the careful and expert work on this manuscript. We have added the suggested text to the legend.

Reviewer #2 (Remarks to the Author):

Thanks for the clarification. The authors have fully addressed my concerns. >> Thank you!